# Reconfigurable, Stretchable Strain Sensor with the Localized Controlling of Substrate Modulus by Two-Phase Liquid Metal Cells

**DOI:** 10.3390/nano12050882

**Published:** 2022-03-07

**Authors:** Linna Mao, Taisong Pan, Junxiong Guo, Yizhen Ke, Jia Zhu, Huanyu Cheng, Yuan Lin

**Affiliations:** 1School of Material and Energy, University of Electronic Science and Technology of China, Chengdu 610054, China; linna_m00@outlook.com; 2School of Electronic Information and Electrical Engineering, Chengdu University, Chengdu 610106, China; guojunxiong@cdu.edu.cn; 3School of Electronic Science and Engineering, University of Electronic Science and Technology of China, Chengdu 610054, China; xysfkyz@126.com; 4Department of Engineering Science and Mechanics, The Pennsylvania State University, University Park, PA 16802, USA; jmz5364@psu.edu (J.Z.); huanyu.cheng@psu.edu (H.C.); 5Medico-Engineering Cooperation on Applied Medicine Research Center, University of Electronics Science and Technology of China, Chengdu 610054, China

**Keywords:** strain sensor, strain engineering, heterogeneous design, two-phase liquid metal

## Abstract

Strain modulation based on the heterogeneous design of soft substrates is an effective method to improve the sensitivity of stretchable resistive strain sensors. In this study, a novel design for reconfigurable strain modulation in the soft substrate with two-phase liquid cells is proposed. The modulatory strain distribution induced by the reversible phase transition of the liquid metal provides reconfigurable strain sensing capabilities with multiple combinations of operating range and sensitivity. The effectiveness of our strategy is validated by theoretical simulations and experiments on a hybrid carbonous film-based resistive strain sensor. The strain sensor can be gradually switched between a highly sensitive one and a wide-range one by selectively controlling the phases of liquid metal in the cell array with a external heating source. The relative change of sensitivity and operating range reaches a maximum of 59% and 44%, respectively. This reversible heterogeneous design shows great potential to facilitate the fabrication of strain sensors and might play a promising role in the future applications of stretchable strain sensors.

## 1. Introduction

Strain sensors, as a signal transducer, can convert external mechanical stimuli into electrical signals. In particular, the stretchable strain sensor is a key part of wearable electronics that can be widely used in applications such as human health monitoring [1,2,3], artificial muscle [4,5], soft robot skin [6,7], and human–machine interfaces [8,9]. Among the electrical-signal-based strain sensors, resistive-type strain sensors consisting of stretchable elastomer substrate and conductive active materials have been extensively investigated [1,10,11], with advantages such as easy-to-setup and high gauge factor (GF). The high performance of such sensors largely relies on the strain transfer from the elastomer substrate to the conductive film, which leads to the crack propagation in active materials and change of conductive path [12,13]. To improve the sensitivity of strain sensors, many previous works have focused on the development of highly conductive materials, such as Ag nanowires [14], graphene [15,16,17,18], carbon nanotubes [19,20,21], and conductive composites [22,23,24,25]. However, the enhancement of sensitivity of the stretchable strain sensor is often accompanied with the degradation of response range.

Recently, strain engineering of elastomer substrate has been employed to regulate the performance of stretchable strain sensors by introducing strain redistribution to the active materials [15,17,26,27,28]. The strain distribution of stretched substrates is affected by the local stiffness based on hooker’s laws and the series and parallel spring model [29]. Consequently, some novel preparation routes of heterogeneous design in the stretchable elastomer substrate have been developed [30,31]. By increasing the local elastic modulus through chemical modifications or introducing mechanical metamaterials, the obtained heterogeneous substrate can significantly enhance the sensitivity of the strain sensors. Although the sensitive performance of the strain sensor could be adjusted by using a heterogeneous strain distribution in elastomer substrate, there are still some limitations regarding the complicated procedure and irreversible programming.

Here, we proposed a heterogeneous substrate design with reversible mechanical modulation by embedding the cells filled with low-melting-point liquid metal (LM) gallium into the elastomer substrate. The phase transition of the LM in the cells enables the local reconfiguration of substrate modulus and redistribution of strain in the soft substrate. The strain sensors based on the controllably heterogeneous substrate show fast response, high stability (1000 tensile cycles under 30% applied strain), and reconfiguration ability, which maximized the relative change in sensitivity and response range up to 59% and 40%, respectively. Endowed with reversibly mechanical tunability by adjusting the phase state of LM in the cells, this heterogeneous substrate design demonstrates promising wide-ranging applicability in stretchable resistive strain sensors, providing considerable insight into scenario-oriented designs.

## 2. Materials and Methods

### 2.1. Materials

Graphene powder and the multi-wall carbon nanotubes (CNTs) were purchased from Suzhou, Tanfeng Tech. (Suzhou, China). Ag flakes and nonionic surfactant (Triton X-100) were purchased from Sigma Aldrich (St. Louis, MO, USA) and used without further modification. PEDOT:PSS (PH 100), purchased from Ossil, Limited. (Sheffield, UK), serves as the electrodes. The SU-8 (2050) photoresist and the developer were purchased from Kayaku Advanced Materials, Inc. (Westborough, MA, USA). Gallium was purchased from Aladdin. Co., Ltd. (Shanghai, China). Polydimethylsiloxane (PDMS, Sylgard 184 Silicone Elastomer Kit) was purchased from Dow Corning (Midland, MI, USA).

### 2.2. Preparation of Graphene/CNTs Hybrid Film

The graphene powder was added into the 10 mL deionized water with a concentration of 2 mg/mL, followed by ultrasonic treatment for 30 min to obtain graphene dispersion. The preparation of CNTs dispersion started from adding 300 mg multi-wall carbon nanotubes powder into the 100 mL deionized water with concentration of 3 mg/mL. After the ultrasonic disintegration treatment for 2.5 h and centrifugation at 3000 rpm for 10 min, the supernatant solution of the CNTs dispersion was collected after standing overnight. The graphene dispersion and CNTs dispersion were mixed at a ratio of 5:2, and the hybrid mixture was ultrasonic-treated for another 15 min.

### 2.3. Preparation of the Heterogeneous Substrate-Based Strain Sensors

The microchannel mold was designed based on the Archimedean spiral pattern and prepared via optical lithography. Firstly, the SU-8 photoresist was spin-coated on a silicon wafer at 1000 rpm for 30 s. After the spin-coating, the pre-designed Archimedean microchannel mold was patterned through an ultra-violet photolithography process. The negative mold was prepared after development process. Subsequently, Silicone elastomer polydimethylsiloxane (PDMS) was prepared by mixing the base and curing agent at a weight ratio of 10:1. The liquid mixture was degassed in a vacuum chamber for 30 min and then poured into the silicone negative mold. The mold filled with the PDMS was spuns at 500 rpm for 10 s to remove excess elastomer, and cured in an oven at 60 °C for 1 h. Another PDMS film was prepared by spin-coating at 500 rpm for 30 s. The flat PDMS film was also cured in an oven at 60 °C for 20 min. Then, the molded PDMS film was bonded on the prepared flat PDMS film. The bonded structure was further encapsulated by spin-coated PDMS and heat-treated at 60 °C for 30 min. The detailed preparation process of the device is shown in Appendix A.

The prepared graphene/CNTs mixture was spray coated on the heterogeneous substrate for 1 h at a certain rate and flow, and a 1.5 μm-thick hybrid film was formed on the surface of the device. The Ag-PEDOT:PSS ink was prepared by mixing the highly conductive PEDOT:PSS solution with the Ag flakes at a weight ratio of 2:1 [32]. The gallium to be filled in the cell was pre-heated to 60 °C before injection. After injecting the gallium into the microchannel, the graphene/CNTs conductive layer was wired with the Ag-PEDOT:PSS ink.

A copper wire (a diameter of 100 μm) was inserted into the gallium and served as the electrodes of the electric heating system (Appendix A), and the copper wires provide a nucleation site for gallium crystals to form at room temperature [33]. Then, an infrared thermal imager (ETS320, FLIR Systems, Inc., Wilsonville, OR, USA) was placed above the strain sensor to monitor the temperature change of the gallium units.

### 2.4. Characterization and Measurements

The morphology and microstructure of the hybrid graphene/carbon nanotubes (Gr/CNTs) film were characterized by scanning electron microscopy (SEM, Phenom Pure). The Raman spectrum of the hybrid carbonous film was acquired with the Raman spectroscopy (Horiba iHR500, wavelength: 532 nm). The cross-section image was obtained by a 3D microscope with a large depth-of-field (Keyence VHX-600). The local mechanical properties of the LM under different states were evaluated by mechanical tensile test. The regulation of the heterogeneity in the stretchable substrate was implemented by an external electric heating system. The loading of the tensile strain was performed using a uniaxial testing machine, and the electrical signals of the strain sensors were recorded using a source measurement unit (Keithley 2400) at room temperature.

### 2.5. Finite Element Analysis

Finite Element Analysis (FEA) was performed using the commercial package ABAQUS 6.14. The polydimethylsiloxane (PDMS) substrate was modeled as an incompressible material with a elastic modulus of 1.0 MPa, and the Poisson’s ratio is approximate 0.5, according to the experimental measurement. The gallium filler was divided into two states: a solid state with elastic modulus of 9.81 GPa and a liquid state with elastic modulus of 132 MPa. The elastomeric substrates were modeled by the hexahedron 3D solid element (C3D8R). The spial cells were modeled by four-node composite shell elements (S4R). The thickness of three layer in the heterogeneous substrate was set to 200 μm (the upper PDMS layer, the gallium cells, and the bottom PDMS layer).

## 3. Results and Discussion

### 3.1. The Modulation of Deformation with The Heterogeneous Substrate

Figure 1a shows the schematic illustration of the reconfigurable strain sensor based on the heterogeneous substrate. The microchannel cell for the LM filler is based on an Archimedes spiral with a radius (*l*) of 4 mm, and the track width (*w*) is 300 μm. Five cells are placed in the substrate, with a distance (*c*) of 15 mm; the total length (*a*) and total width (*b*) of the sensor are 80 mm and 20 mm respectively. The detailed structure of the sensor with a heterogeneous substrate is shown in Figure 1b, which includes a sensitive layer (a hybrid Gr/CNTs film) on the top; the middle and bottom layer are PDMS layers to construct the microchannel cells with LM filled. Graphene and CNTs as typical carbon nanomaterials possess high conductivity and have been utilized extensively as active materials. The Gr/CNTs hybrid film, whose resistance changes when cracks appear with the applied strain, is employed as active material since the hybridization of graphene and CNTs is expected to significantly enhance isotropic mechanical properties and strain sensitivity with a cyclically large strain [34,35]. The Raman characterization result of the hybrid Gr/CNTs film is shown in Figure 1c. There are two distinct peaks at approximately 1350 cm${}^{-1}$ and 1580 cm${}^{-1}$, which are attributed to D and G bands, respectively. The thickness of each layer obtained by optical microscopy is shown in Appendix A.

Figure 1d shows the deformational behavior of the cell in the heterogeneous substrate with LMs in solid and liquid phases when applying 22% tensile strain, respectively. When the solid cell is stretched, an obvious difference in the change of distance between the tracks can be noted. A dependence of track distance on the location can be noticed in the cell with solid LM, while the cell with liquid LM deforms uniformly with the same track distance in several locations. As the two-phase gallium LM has a low melting point at 29.76 °C, the LM displays significant stiffness change between liquid phase (above the melting point) and solid phase (below the melting point) with a fast (less than 1 s) phase change transition time [36]. Gallium possesses high stiffness with a high elastic modulus of 9.8 GPa in the solid phase, and with a low modulus of 132 MPa in the liquid phase, while the elastic modulus of the PDMS matrix is 1 MPa [37]. Therefore, the local strain gradually increases along the center track toward the stretching direction when the LM in the cell is in solid phase, leading to the non-uniform deformation of the substrate. Spiral line-shaped channel design results in the inner track performing less than the deformation of the outer tracks when the solid cell is in external tensile. The high-elastic modulus difference in the microchannel cell further contributes to this deformational behavior. The heterogeneous strain distribution on the substrate stems from the synergy effect of the high modulus difference and the structure design of spiral microchannels. Comparatively, the cell filled with LM has a much smaller local elastic modulus and thus results in a homogeneous deformation of the substrate. The effective regulation of substrate deformation by the phase transition of LM provides the opportunity to modulate the crack propagation of the hybrid Gr/CNTs film during the strain detection.

### 3.2. Morphological Properties of the Active Materials

Regarding the sensing mechanism of the resistive-type strain sensors, cracks occur and spread in active sensing films coated over the surface of polymer substrates during the stretching-releasing process, resulting from the mechanical mismatch with the supporting materials [11,38]. As shown in Appendix A, the Gr/CNTs hybrid film was stacked with a graphene film and a CNTs film. The dense graphene network was interspersed with a large number of CNTs, which formed the conductive network. When the device is stretched, cracks appear at the connection nodes of the graphene flakes, which increase the resistance of the hybrid conductive film. The morphology of the Gr/CNTs film was further observed with SEM, and the results in Appendix A show that the CNTs and graphene flakes were welded into a complex collective. The existence of the graphene enhanced the tensile strength of the CNTs film. After the cyclic tensile test, it can still be observed that the graphene and CNTs at the cracks are intertwined and connected (Appendix A). Cracks tend to expand in the stress-concentrated areas. The rapid separation of nanomaterials at the microcracks edges leads to a significant increase in the electrical resistance of strain sensors and further contributes a higher sensitivity with the applied strain. To further illustrate the dependence of the crack distribution on the applied strain, we thereby observed the crack distribution using SEM. Figure 2 shows the SEM images of surface morphologies of the hybrid Gr/CNTs film on the heterogeneous substrates with 40% applied strain when the LM in all the cells is in solid or liquid phases. Three regions, which are labeled as yellow, are picked from the inside, intermediate, and periphery tracks for the characterization. When the LM in the cell is in the solid phase, the cells are expected to feature a significant modulus difference from the PDMS, which leads to an uneven strain distribution in the hybrid Gr/CNTs film. As shown in Figure 2a, cracks in the hybrid Gr/CNTs films can be clearly observed when the sensor with solid-phase LM is stretched. The distance between cracks gradually decreases from the center of spiral tracks to the outer edge (d1 > d2 > d3), and the crack density in the outer edge notably increases. The uneven distribution of cracks, together with the dependence of crack distance on the location, indicates the inhomogeneous modulus of the substrate with the solid LM can effectively modulate the interaction between the active material and substrate by regulating the deformation of the substrate. Figure 2b illustrates the distribution of cracks in the sensor with LM in liquid phase. No significant difference of crack density is observed in the same three regions, indicating a uniform crack distribution in the hybrid Gr/CNTs film. The consistency between a crack distribution and cell deformation shows the effectiveness of modifying the response of the active material to the applied tensile strain by configuring the modulus of the strain sensor’s substrate.

### 3.3. FEA Analysis of The Strain Distribution in the Heterogeneous Substrate

The modulation of the strain by the phase transition of LM is further investigated by FEA. In the simulations, the sensor is stretched with 30% applied strain along the length direction of the sensor, and other parameters are provided in the Appendix A. Figure 3a shows the strain in the sensor when LM in all the cells is in solid phase or liquid phase. Obvious contrast of the size of strain between the cell region and the rest of the substrate appears when the LM in the cells becomes solid. Moreover, the maximum achievable local strain doubles with the phase transition. To further illustrate strain distribution in the cell, a path is selected from the center to the periphery of cell tracks along the stretching direction, which is noted in the right panel of Figure 3a. Figure 3b shows the local strain distribution along the path when the LM in all cells is in the solid phase. The strain gradually increases along the path, and the local strain in the periphery track is magnified to be greater than the applied strain. Comparatively, the local strain is redistributed uniformly along the path in the all-liquid case and the size of the strain is kept below the applied strain (Figure 3c). Moreover, by comparing the strain distribution of the same region under different heterogeneity in the substrate (Appendix A), the results show that the overall heterogeneity change in substrate again affects the local strain distribution. Generally, the FEM results theoretically demonstrate the flexibility of regulating the strain distribution of the heterogeneous substrate via the phase transition of LM in the cells.

### 3.4. Resistance Response Properties of the Stretchable Strain Sensor with Heterogeneous Substrate

In Figure 4, the electrical performance of the device are illustrated. Figure 4a shows the dependence of the resistance changes ($\Delta R/R_{0}$) of the proposed strain sensor on the applied strain when the LM of all the cells is in solid phase or liquid phase. Higher sensitivity is obtained in the solid-phase case, while the liquid-phase case results in a larger response range but lower sensitivity. Figure 4b shows the resistance change of the sensor with applied strain up to 30% when introducing the phase transition to selected cells with the electric-heating control system (Appendix A). The resistance-strain relation for the applied strain ranging from 0% to 30% exhibits high linearity with the Pearson correlation coefficient as high as 0.997. The corresponding fitting result is shown in Appendix A. Significant relevance between the number of “liquid cells” in the substrate and the sensor’s response performance can be observed, indicating the possibility to obtain the multiple combinations of sensitivity and response range by realizing the partial phase transition of LM in several cells. Figure 4c further shows the relation between the sensor’s gauge factor (GF) and phase transition of LM cells. GF is defined as the ratio of relative resistance ($\Delta R/R_{0}$) and applied strain (ε): $\mathrm{GF}\;=\;\frac{\Delta R/R_{0}}{\varepsilon }$. When the phase transition state gradually undergoes towards “all liquid state”, the heterogeneity of the substrate decreases, resulting in the decrease of GF accordingly. The GF can be regulated from 220 in all-solid-state to 90 in all-liquid-state. Cyclic tests are also conducted with the proposed sensor. Appendix A illustrates the relative resistance change variation of strain sensors maintain steady under stretching/releasing cycles for various strains, indicating its superior stretchability and stability. Furthermore, as shown in Figure 4d, the sensor’s response to the applied strain can be dynamically and reversibly tuned during the cyclic test. The amplitude of the resistance change varies in accordance with the phase transition state even when the tensile strain is repeatedly applied. As shown in the Appendix A, the conventional strain sensors largely depends on the intrinsic properties of the active material, making it difficult to solve the dilemma between the sensitivity and operation range. Therefore, the programmable soft substrate with an adjusted elastic modulus provides an effective implementation method for strain sensor with a controllable sensing characteristic, which allows the strain sensor to have multiple sensing modes simultaneously. Appendix A shows the good dynamic response performance of device with the heterogeneous substrate in a linear sensing range of 0–30%. The hysteresis performances under various states are shown in the Appendix A. Moreover, as indicated by Figure 4e, the strain sensor retained high performance after 1000 cycles with the periodical loading of 30% applied strain, which indicates high stability of the sensor with the heterogeneous substrate design.

## 4. Conclusions

In summary, a reconfigurable and stretchable strain sensor with multiple combinations of sensitivity and response range is realized by a novel heterogeneous substrate design. The introduction of the spiral cell array filled with two-phase liquid metal enables the local modulus modulation in the substrate and leads to strain redistribution in the active material. The stretchable strain sensor with the heterogeneous substrate design is experimentally fabricated with the Gr/CNTs film as the active material. Theoretical and experimental results verify the ability of the proposed design to regulate the sensitivity and response range of the sensor dynamically and reversibly by controlling the phase transition state of the LM in the spiral cells. The GF and response range of the sensor can be tuned from 220 to 90 and 25% to 45% when the solid–liquid phase transition happens for the LM in all cells in the substrate. The demonstrated features of the controllable heterogeneous design strategy via introducing a phase-changed liquid metal unit offer a promising route for the smart fabrication of future soft robotics and smart systems.

## Figures and Tables

**Figure 1 nanomaterials-12-00882-f001:**
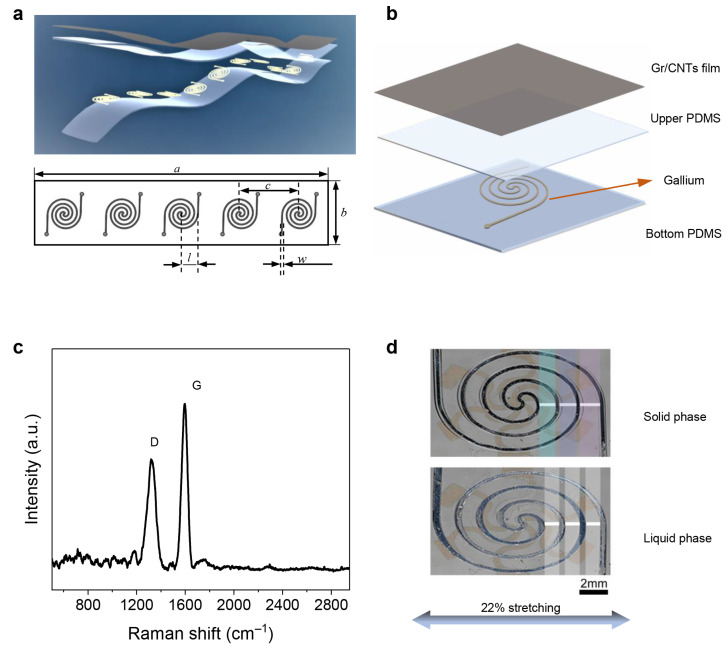
(**a**) Schematic illustration of the strain sensor based on a heterogeneous substrate with cell array. (**b**) The exploded view of the detailed structure of the sensor with a heterogeneous substrate. (**c**) Raman spectrum of the hybrid Gr/CNTs film. (**d**) The local deformation of the cell filled with LM in solid and liquid phases.

**Figure 2 nanomaterials-12-00882-f002:**
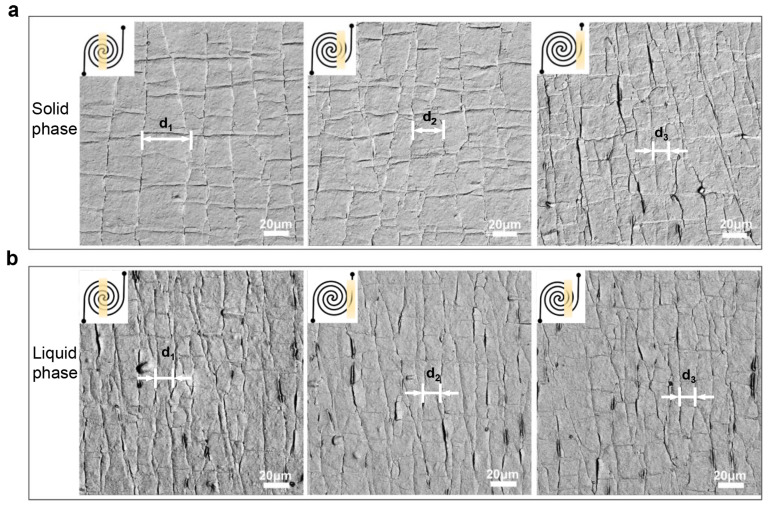
The SEM image of surface conductive materials (the Gr/CNTs hybrid film) on the gradient parts in a heterogeneous substrate-based strain sensor (*l* = 4 mm, *w* = 250 μm) with the 40% applied strain. The yellow labels correspond to the locations of the cracks. The gallium in cells is in: (**a**) all solid phase and (**b**) all liquid phase.

**Figure 3 nanomaterials-12-00882-f003:**
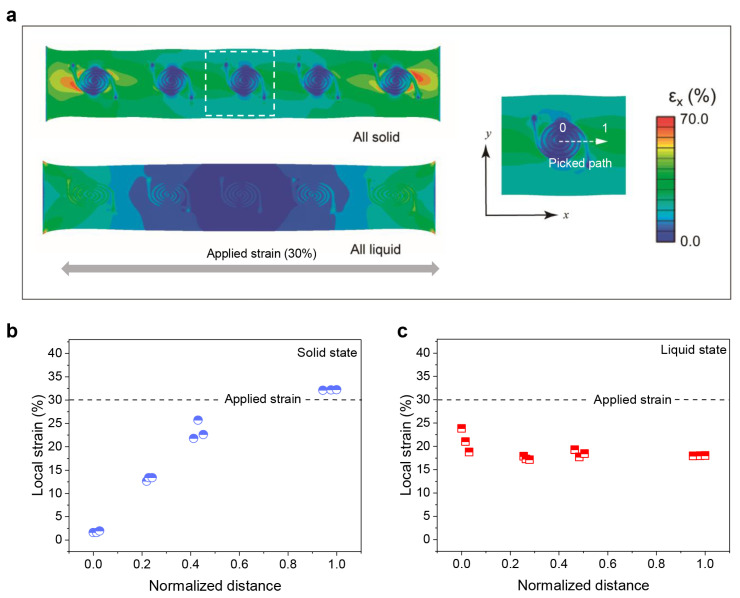
The FEM results of the strain distribution in the stretchable strain sensor with the heterogeneous substrate design. (**a**) The overall strain distribution in the sensor when the LM in the cells is all in solid phase or liquid phase (**left**). The zoomed-in strain distribution in the cell region with solid LM (**right**). The dotted box indicates the picked path for profiling of local strain distribution. (**b**,**c**) Strain distribution along the picked path when the LM in all cells is in solid phase (**b**) and liquid phase (**c**).

**Figure 4 nanomaterials-12-00882-f004:**
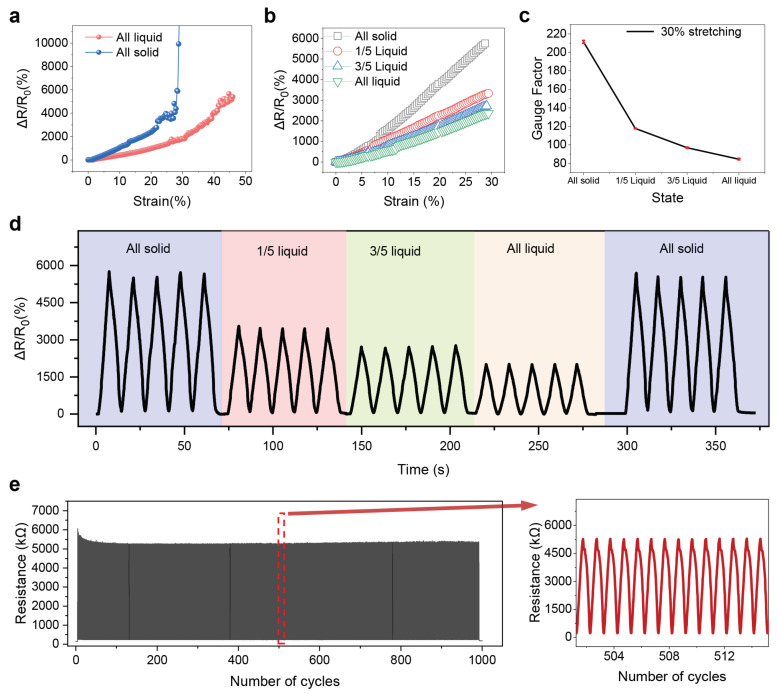
The electrical performance of the strain distribution in the stretchable strain sensor with the heterogeneous substrate design. (**a**) Relative resistance changes as a function of applied strain when the LM in the cell are all in solid phase or liquid phase. (**b**) Relative resistance changes as a function of applied strain when the LM in selected cells is in liquid phase. (**c**) The dependence of gauge factor on the phase transition state. (**d**) The relative resistance change recorded in the cyclic test when the sensor is stretched with 30% applied strain and the number of “liquid cell” is dynamically tuned. (**e**) The relative resistance change recorded in the cyclic test when the sensor is stretched with 30% applied strain and the LM in all cells are in solid phase.

## Data Availability

The data presented in this study are available on request from the corresponding author.

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
