# Peer review of "Reconfigurable, Stretchable Strain Sensor with the Localized Controlling of Substrate Modulus by Two-Phase Liquid Metal Cells"

_nanomaterials, 2022, doi:10.3390/nano12050882_

Round 1

Reviewer 1 Report

Paper discusses an interesting scientific topic, proposing an innovative stretchable strain sensor.

Author Response

We thank the reviewer for recognizing the novelty of this work.

Reviewer 2 Report

In this work, Lin et al. reported strain sensors based on reversible phase transition of the liquid metal. Some characterization of the device was conducted. Part sensing properties of the device were tested and described. The structure of the device was relatively simple and the method adopted here was facile. Please consider the following suggestions.

(1) How does the sensor work and what is its sensing mechanism? Also, what is the structure? The authors need to provide a sensing schematic of the device so that more readers will know it better.

(2) In Figure 4a, the change in resistance to the strain was not linear, as lease in two sections. The authors should describe the GF with corresponding strain range. In addition, sensing range is also an important parameter of strain sensor. What are the sensing ranges of the devices?

(3) Many sensing mechanisms, functional materials and methods have been recently reported for the fabrication of strain sensor, including but not limited to DOI: 10.1002/aisy.202100193, DOI: 10.1002/adfm.202101107, DOI: 10.1039/c7mh00071e, DOI: 10.1039/c8tc02655f, and DOI: 10.1021/acsami.8b20245. What are the different between them and this work, such as methods, the advantages and disadvantages of performance? Some discussion may be added so that potential readers would know the development of strain sensor better.

(4) The references format needs to be modified according to the requirements of the journal.

(5) How about the signal stability of the sensor under different strains, such as 5%, 10%, 15%, 20% 25%, and 30%? A stable signal is important to well detect strain. The authors need to give experimental data.

(6) How are the electrode wires of the device fixed and led out? Does the stretchability affect the connecting between the electrodes and the composite?

Reviewer 3 Report

Dear Authors,

I compliment for the paper and the research idea, very innovative,

still I think there is some minor revision to do in the presentation of the results.

Let's begin with some minor spell error:

Line 98 Missing capital letter

Line 139 Missing space from caption

Line 248 Thermo-optical

The minor revision about results presentation regards Fig. 4

Fig. 4e is nice but it should be moved upward, just after

Fig. 4c. Furthermore, I think it is right to make it complete, I mean,

after all liquid coming back to 3/5 and 1/5 phase, so to put in figure 4c

also the coming back strain gague values.

What it is really useless it is figure 4d that does not give any information.

It is enough to cite in the text at line 225. 

What it is missing ? A graph of the Sensor behavior in the hysteresis form

that will evidence the precision of the Sensor itself. From Fig. 4e it is so 

evident that this type of Sensor has problems in the hysteresis curve. Please 

add figure and relative analysis to the text.

Reviewer 4 Report

In this manuscript, the authors developed a novel design for reconfigurable strain modulation in the soft substrate with two-phase liquid cells. The relative change of sensitivity and operating range reaches a maximum of 59% and 44% respectively. The analysis made in the manuscript is clear for understanding the subject. This work is recommended to be published before the following observations are addressed.

  • It hard to observe the difference between two graphs at solid and liquid phases in Figure 1d and scale bars should be added.
  • In general, the heat induced by the liquid metal may also influence the property of Gr/CNT. Its resistance may be increased or decreased by changing the surrounding temperature. Therefore, the authors are encouraged to investigate the temperature-dependent resistance change of Gr/CNT.
  • To realize high-performance strain sensors, the authors may refer to related publications: 1. Journal of Materials Chemistry C, 2019, 7(31): 9609-9617. 2. Advanced Materials, 2021, 33(18): 2008701. 3. ACS Applied Materials & Interfaces, 2020, 12(37): 42420-42429.

Round 2

Reviewer 2 Report

Most of my concerns have been well addressed by the authors. I have no other suggestions.